# Calculating and comparing codon usage values in rare disease genes highlights codon clustering with disease-and tissue- specific hierarchy

Rachele Rossi[1,2☯], Mingyan Fang[3☯], Lin Zhu[3,4], Chongyi Jiang[3], Cong Yu[5], Cristina Flesia[6], Chao Nie[3], Wenyan Li[7], Alessandra Ferlini[1,2]*

1 Unit of Medical Genetics, Department of Medical Sciences, University of Ferrara, Ferrara, Italy, 2 Dubowitz Neuromuscular Unit, Institute of Child Health, University College London, London, United Kingdom, 3 BGI-Shenzhen, Shenzhen, China, 4 BGI College & Henan Institute of Medical and Pharmaceutical Sciences, Zhengzhou University, Zhengzhou, China, 5 BGI Genomics, BGI-Shenzhen, Shenzhen, China, 6 Department of Earth and Environment Science, University of Milano-Bicocca, Milano, Italy, 7 MGI, BGI-Shenzhen, Shenzhen, China

☯ These authors contributed equally to this work.
* fla@unife.it

**Data Availability Statement:** All relevant data are within the paper and its Supporting information files.

## Abstract

We designed a novel strategy to define codon usage bias (CUB) in 6 specific small cohorts of human genes. We calculated codon usage (CU) values in 29 non-disease-causing (NDC) and 31 disease-causing (DC) human genes which are highly expressed in 3 distinct tissues, kidney, muscle, and skin. We applied our strategy to the same selected genes annotated in 15 mammalian species. We obtained CUB hierarchical clusters for each gene cohort which showed tissue-specific and disease-specific CUB fingerprints. We showed that DC genes (especially those expressed in muscle) display a low CUB, well recognizable in codon hierarchical clustering. We defined the extremely biased codons as "zero codons" and found that their number is significantly higher in all DC genes, all tissues, and that this trend is conserved across mammals. Based on this calculation in different gene cohorts, we identified 5 codons which are more differentially used across genes and mammals, underlining that some genes have favorite synonymous codons in use. Since of the muscle genes clear clusters, and, among these, dystrophin gene *surprisingly* does not show any "zero codon" we adopted a novel approach to study CUB, we called "mapping-on-codons". We positioned 2828 dystrophin missense and non-sense pathogenic variations on their respective codon, highlighting that its frequency and occurrence is not dependent on the CU values. We conclude our strategy consents to identify a hierarchical clustering of CU values in a gene cohort-specific fingerprints, with recognizable trend across mammals. In DC muscle genes also a disease-related finger-print can be observed, allowing discrimination between DC and NDC genes. We propose that using our strategy which studies CU in specific gene cohorts, as rare disease genes, and tissue specific genes, may provide novel information about the CUB role in human

**Funding:** The SOLVE-RD (grant agreement n. 779257) H2020 EU projects (to AF as full partner) and the National Natural Science Foundation of China (No. 31800765, to MF) are acknowledged The funders had no role in study design, data collection and analysis, decision to publish, or preparation of the manuscript and the authors received no specific funding from them for this work, but Authors wish to acknowledge them for their support of AF and MF omics researches.

**Competing interests:** The authors have declared that no competing interests exist.

and medical genetics, with implications on synonymous variations interpretation and codon optimization algorithms.

## Introduction

The genetic code consists of 64 triplet codons encoding 20 amino acids and three stop codons, these last being recognized by the translational machinery to interrupt the protein synthesis [1]. With the exception of two amino acids, tryptophan and methionine, which are encoded by a unique codon, all other amino acids recognize multiple synonymous codons based on two, three, four or six triplet redundancies, a phenomenon known as codon degeneration. There is intriguing evidence that the redundancy of genetic code played a crucial evolutionary role in allowing protein synthesis to turn the RNA world into the protein world [2]. For reasons not fully understood, some codons become poorly used, a phenomenon known as codon usage bias (CUB), or even tend to disappear (extreme CUB) during evolution. Although CUB was widely studied in various gene categories (via gene ontology or interactome maps) and across species, its evolutionary meaning is still uncertain [3–5]. The original neutral theory of molecular evolution [6] might not apply to codon selection and mutational pressure and natural selection may have played a major role in contributing to the codon usage [7]. Indeed, although only in a few reported cases such as for keratin and some ribosomal and mitochondrial genes, an extreme CUB was identified in human, chimpanzee and chicken implying a constant bias within vertebrates, and suggesting its evolutionary meaning [4].

A large consensus exists about the concept that the choice of a synonymous codon does affect protein translation efficiency, expression level, structure, and function, a notion that has prompted the designation of optimal codons and codon optimization, which is a process routinely used in synthetic biology to increase protein expression [8]. Nevertheless, there is little consensus among the various codon optimization algorithms, and the metrics currently used might not be appropriate for all genes [8].

The meaning of synonymous variations in the human genome and their effect on hereditary diseases is largely unknown [9, 10]. Interpreting their functional impact on genes is difficult, if not impossible, without dedicated functional assays. In silico tools are currently used to decipher synonymous variations but they are inaccurate. In addition, synonymous changes are fully disregarded and filtered out of genomic data outputs, a fact that causes their omission in pathogenic variation discovery and validation and lack of potential novel disease genes or the identification of pathogenic genotypes.

In terms of energy, an extreme CUB is predicted in keeping energy demand low for protein translation, according to the maximum entropy principle, which may drive the progressive increase of CUB during evolution and across species [11, 12]. This trend highlights certain functional pathways that might have been given energetic priority (assuming bias towards preferred codons) via natural selection and may have occurred in gene families with particular relevance in a given lineage [13]. For example, in genomes with high GC content (such as *Homo sapiens*, *HSA*), which can prime SNP changes, extreme CUB frequently occurs and it is thought to reduce the risk of nonsense variation occurrence [14–16].

The role of "rare" (extremely biased) codons through evolution is still controversial. In bacteria, rare codons near the 5' end of genes facilitate removing translation repression and are thought to be "highway on-ramps" to prime and accelerate protein translation of 60 times more, with a key role in regulating the ribosomal trafficking [17]. Synergistically, folding

properties of mRNA rich in rare codons at their 5' end increase the translation speed, like in rapidly dividing cells [18, 19]. Conversely, some frequently used codons have the opposite effect, slowing down translation efficiency [20].

We studied CU values and profiled CUB in 31 tissue-specific, muscle, kidney and skin, human genes whose mutations do cause rare diseases (disease-causing genes, DC) and compared them to 29 human genes which are not related to human diseases (Non-Disease-Causing genes, NDC) but have the same tissue expression. CU values were calculated also across 15 mammalian species to explore whether CU values may vary across the selected human genes with a consistent evolutionary trend. We showed that CUB displays a tissue-specific codon fingerprint which is also different in DC compared to NDC genes.

We also explored relationship between CUB and mutations in the muscle dystrophin (*DMD*) gene, the larger X-linked human gene, which mutations do cause Duchenne muscular dystrophy [21] and highlighted that its 2828 pathogenic variations occurred non-randomly in codons.

We suggest that our human gene disease-driven approach may help in identifying critical codons, which may play a role in gene mutational events, synonymous variation interpretation and possibly codon optimization algorithm design.

## Methods

Our strategy was based on comparing CU values and their hierarchical clusters in *Homo Sapiens* muscle, kidney, and skin tissues, across the selected mammals, and in DC versus NDC genes.

### Species selection, sequence data and sources for CU values calculation

We selected the following 15 mammals in the metazoan phylogenetic tree: *R. ferrumequinum* (Greater horseshoe bat), *M. musculus* (mouse), *F. catus* (cat), *C. lupus familiaris* (dog), *E. caballus* (horse), *B. Taurus* (cattle), *M. murinus* (gray mouse lemur), *G. variegatus* (Sunda colugo), *C. jacchus* (common marmoset), *M. mulatta* (macaque), *N. leucogenys* (gibbons), *P. abelii* (orangutan), *G. gorilla* (gorillas), *P. troglodytes* (chimpanzee) *and H. sapiens* (human).

S1 Table shows the list of species and assembly genome sequences. The mRNA reference sequences of all *H. sapiens* gene groups were retrieved from RefSeq and GeneBank at the National Center for Biotechnology Information [22] as shown in S2–S4 Tables.

Mammals were selected based on the higher number of annotated genes (RefSeq in S5–S10 Tables).

We selected genes with different lengths such as mRNAs and, to maximize the number of analyzed codons and avoid bias against short or partial sequences, only full-length coding sequences were selected. If more than one splicing isoform was annotated, the longest isoforms were selected, since recent single cell RNAseq studies showed that mRNA length does not influence isoform tissue expression level [23].

### Gene selection

DC genes were prioritized based on their involvement in Mendelian rare diseases, with incidence less than 1:5000 (according to the OMIM catalogue, www.omim.org), and with a homogenous and tissue/organ-specific-related phenotype (renal, skeletal muscle and skin). Exclusion criteria for genes were involvement in polygenic or cancer (both Mendelian and somatic) diseases and absence of tissue specificity (housekeeping genes). We then selected DC genes based on their disease-causing and tissue-specificity and NDC genes based on their

higher expression in same tissues where DC genes were also expressed (muscle, skin and kidney).

Gene selection criteria (S2–S4 Tables) were based on: i) fully annotated genes known to cause rare Mendelian diseases (DC genes), ii) fully annotated genes not causing rare Mendelian diseases (NDC genes).

The OMIM catalogue database [24] was used to categorize genes as DC or NDC. The DC genes had to be associated with a Mendelian disease in one of three tissues selected (kidney, skeletal muscle and skin) in at least 5 reported families/patients, therefore being confirmed in the OMIM database, and with defined inherited patterns (autosomal recessive, autosomal dominant or X-linked recessive). Evidence of gene involvement in both Mendelian and somatic cancers, as susceptibility genes, were considered exclusion criteria for gene selection, since cancer Mendelian genes are often cancer predisposing genes as well, and then they might be confounding factors in our study which is focused on Mendelian diseases only. Rare disease (RD) types are related to muscle, skin and kidney tissues gene mutations. Muscle RDs include muscular dystrophies and congenital myopathies, which causative genes are predominantly and highly expressed in skeletal muscle such as dystrophin, dysferlin, or in muscle extracellular matrix, as the collagen 6A1 gene. Kidney RDs include uromodulin and polycistin 1 gene, whose mutations cause Tubulointerstitial kidney disease or polycystic Kidney Type 1 respectively, gene highly expressed in these two different kidney compartments. Finally, skin RDs include keratin 10 and HOXC13 genes, whose mutations are associated with epidermolytic hyperkeratosis and ectodermal dysplasia 9, two different diseases in terms of phenotype and skin layer involvement. S2–S4 Tables, report the full list of RD genes with all corresponding OMIM numbers.

Two genes, *TMEM52B* and *PLA2G4E*, not listed in the OMIM database since they have never been associated with any human diseases, were checked using PubMed [25], ClinVar [26] and DMDM [27] databases and thus excluded from being causing Mendelian diseases.

We selected tissues based on their high gene enrichment in the Human Protein Atlas (HPA) database [28]. Our gene prioritization was based on HPA metrics used for RNA level (Transcripts Per Million, TPM), protein expression score (high, medium, low, not) and tissue-specificity values. These scores allowed us to classify NDC and DC gene expression profiles according to their tissue specificity. The higher expression value implies at least four-fold higher mRNA levels in selected tissues compared to any other tissues, while protein scores were high or medium, low levels of expression were excluded.

## Mutation databases

We examined the OMIM [24], ExAC [29] and ClinVar [26] public databases together with our UNIFE internal databases [21] for *DMD* single nucleotide gene variations. Only proven, pathogenic missense and nonsense variations were considered, since their meaning and identification in the databases are not equivocal [30]. Synonymous changes in the *DMD* gene were therefore not considered in this study since they are almost invariably defined as variants of uncertain significance (VUS) or benign variants, according to the ACMGG guidelines [31].

## CU values calculation and statistical analysis

The codon usage (CU) frequency was independently calculated for each of the 19 amino acids considered in each gene group. The three stop codons usage was also evaluated. Methionine (AUG) and tryptophan (UGG) were not included since they are encoded by a unique triplet. All statistical analyses were conducted with R-3.4.4 [32].

Statistical significance was defined as P value < 0.05. Basic statistical tests and generation of bar plots and box plots were performed by using built-in functions included with the base

distribution of R or functions in ggplot2 package [33]. For codon usage comparison between DC and NDC genes, tissue types and species, a Wilcox on rank sum test was applied to calculate two-tailed P values using the 'Wilcox test' function in R [34], and they were visualized by using R package ggplot2.

CU frequency in DC and NDC genes was also compared to identify the codons most differently used in gene tissues, across species and in DC versus NDC genes, and data were visualized in box plots.

The Spearman correlation coefficient of codon usage frequency in *HSA* muscle, skin and kidney DC and NDC genes was used and was visualized using R package ggplot2.

Function heatmap.2 in R package gplots was used for the clustering of codons and their visualization. In the clustering, as the codon usages are interval data and are not affected by outliers with extremely large values, Euclidean distance metric was selected for easy implementation and simple interpretation. Agglomerative hierarchical clustering was performed by using the default "complete" method in function hclust.

Genes are always listed in graphs in decreasing order according to the number of exons, and exon number was calculated based on annotation data downloaded from the Ensembl Genome database [35]. Data were analyzed using the hierarchical cluster algorithm or applying priority analysis in terms of CUB percentages.

The synonymous codons which are not used at all by genes, therefore having an extreme codon bias, were named "zero-codons".

## Results

S5–S10 Tables report CU values calculated in muscle DC and NDC, skin DC and NDC, and kidney DC and NDC genes, respectively. On these values we carried on CU values verification and comparison in the above listed cohorts.

### *Homo sapiens* (*HSA*) genes tissue-specific CUB fingerprint

We firstly verified the hierarchical clustering of synonymous codon usage in all studied genes with different tissue specificity (muscle, skin, and kidney) in *HSA*. We observed a tissue-specific codon clustering, which we defined "CUB fingerprint". Heat plot graphs (Fig 1) show that clustering of frequently used (in red) and rarely used (dark blue) codons greatly vary among human tissues. In muscle genes, extremely biased codons (low CU values, blue key color, or high CU values, dark red key color) are tightly clustered in terms of both gene and codon types, while intermediate CU values (light blue to yellow key color) are dispersed in the trees (Fig 1A).

The CUB fingerprint of skin genes (Fig 1B) is characterized by predominant low CU values (yellow dots) with a few dispersed red dots and a clear clustering of rarer (dark blue dots) codons, while the few codons with intermediate or high CU values are unevenly distributed among genes with high distances in the tree. Gene-related codon clusters are also less defined compared to muscle and kidney (see below). In kidney genes, the CUB fingerprint differs from the other two gene groups and the codon hierarchy is more defined (only two main lineages, Fig 1C). Codons with intermediate values are clustered with gene-related trends, as visible in *PKD2*, *KCNJ1* and *MIOX genes.* The vast majority of genes have low or intermediate CU values with wide and diffuse blue spot clusters. A small group of genes (UMOD, BSND, SLC22A8, MIOX, AQP6, PKD1, SLC12A3, and GGACT) show high CU values of codon UGA, UAC, UUC, AUC, GAC, AAC, AAG, GAG, UGC, CAC, and CAG, belonging to well defined lineages in the cluster (gene-codon specific CUB fingerprint).

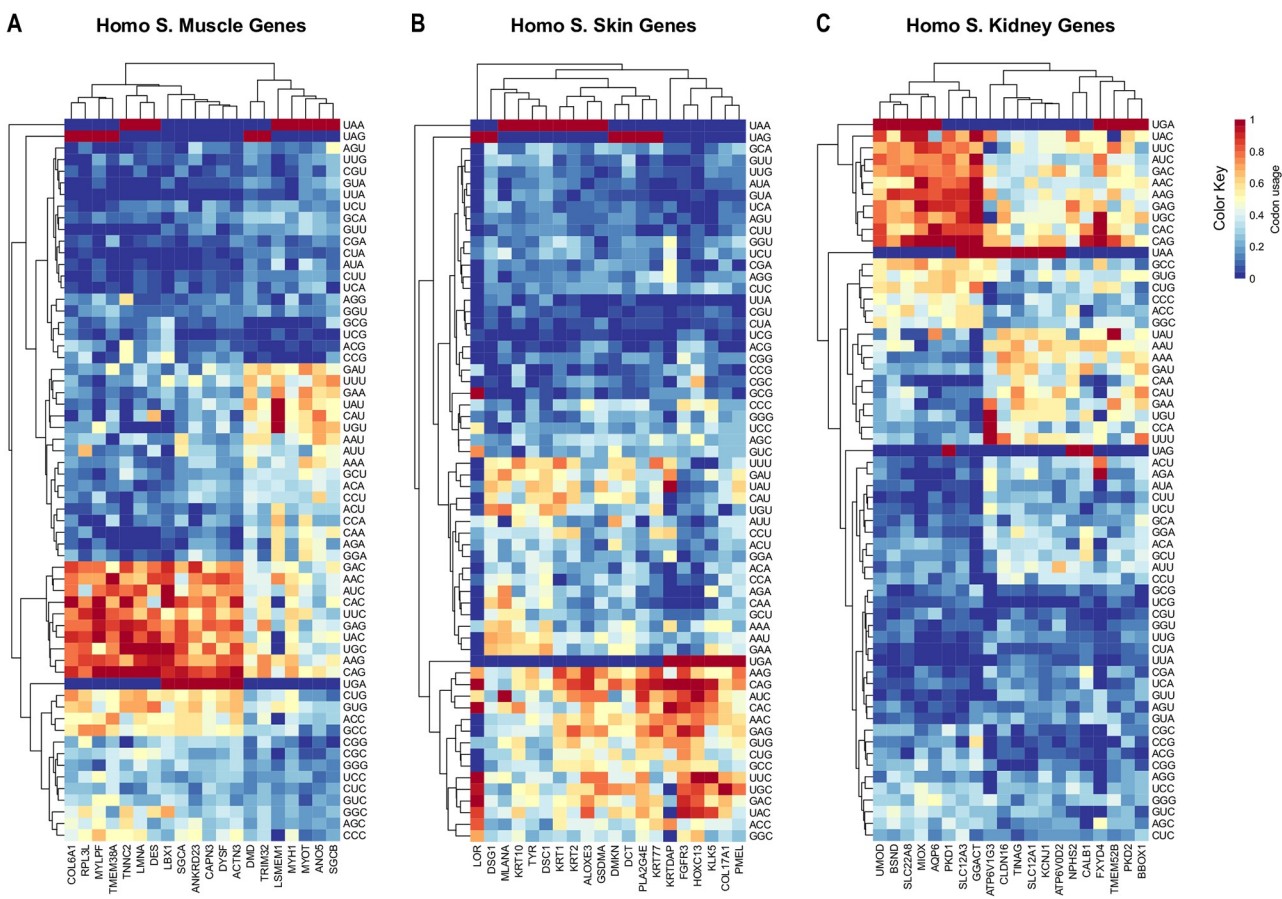

**Fig 1. CU values in *Homo sapiens* muscle, skin, and kidney genes.** Heat plots were generated using R package gplots. Rows were clustered based on Euclidean distance. The color coding varies from dark blue to red with low to high CU values, respectively. Hierarchical clustering of synonymous codon usage in all studied genes in different tissues (muscle, skin and kidney) in *HSA* was generated. The heat plot graphs show that the clustering of frequently used codons (in red) and rarely used codons (dark blue) greatly varies among genes and tissues. In muscle genes, extremely biased codons (low CU values, key color dark blue, or high CU values, key color dark red) are tightly clustered in terms of both gene and codon types, while intermediate CU values (key color light blue to yellow) are more dispersed in the trees (panel A). Among muscle genes, only *DMD* does not show any extreme CUB, since no red spots (corresponding to higher CU values) occur (Fig 1A). The CUB fingerprint of skin genes (panel B) shows a prevalent low CU values codons (dark blue), with a few dispersed, not clustered, inhomogeneous distributed yellow-red spots (intermediate and high CU values). CU clusters are less defined compared to muscle. In kidney genes (panel C), the CUB fingerprint also differs from the other two gene groups. The vast majority of genes have low or intermediate CU values (wide and diffuse yellow-blue spot clusters) with a cluster of high CU values closely clustered and related to UMOD, BSND, SLC22A8, MIOX, AQP6, PKD1, SLC12A3, and GGACT genes. Interestingly all of these genes are DC.

Looking at the codon clustering, the most used codons, CAG, AAG, UGC, GAG, UAC, UUC, CAC, AAC, AUC and GAC, have an identical clustering and overlap in muscle and kidney but not in skin genes (Fig 1A and 1C, left sides). Among muscle genes, only *DMD* does not show any extreme CUB, since no red spots occur (Fig 1A), with the only obvious exception of the unique stop codon UAG.

Looking at the gene clustering, muscle and kidney gene dendrograms show recognizable fingerprints (vertical hierarchy in the heatmaps in Fig 1A and 1C), since genes with high or low CU values (enriched with red or blue spots) are clearly clustered (13 genes in Fig 1A, and 8 genes in Fig 1C, left sides), while in skin genes this does not occur. Therefore, as noted above, some gene-specific CU values were observed, depending on the tissue studied.

Stop codon usage values were also calculated in the three gene groups. In skin and muscle genes, all stop codons are uniformly used, with UAA more frequently present. In kidney genes, UAG is very rarely used, while UAA and UGA are equally represented.

## CUB fingerprint across mammals

We analyzed CU values in all 20 tissue-specific genes across 15 mammalian species in the metazoan phylogenetic tree (Fig 2A–2C) or CU values in all mammals and among the three gene tissues (Fig 2D–2F). CU values across mammals show evident tissue-specific CUB fingerprints due to different codon type usage (Fig 2A–2C).

CAG, AAG, CAC, GAC, GAG, AUC, AAC, UAC, UGC and UUC are the most frequently used codons (red spots) in all gene tissues and across mammals, while UUA, CUA, UCG, CGU, CUU, GUA, CGA, AUA, UCA, UUG and GCG are the rarest codons (blue spots) in muscle and skin but not in kidney genes. Muscle and skin CUBs do have a similar behavior of CU values clustering, although with lower CU values in skin (more yellow spots), similar to kidney genes which show a different hierarchical clustering.

Looking at the gene clusters in all mammals (Fig 2D–2F), the CU values clusters cannot be seen. All tissues have a different CUB fingerprint, also with different codon hierarchical clustering. Gene-related dendrograms are also different since muscle (13/20), kidney (8/20) and skin (4/20) genes cluster together with different CU values related to different codon types. Considering all genes in all species codon types vary indeed. Therefore, although some tissue-related CUB fingerprints are still recognizable in mammals, no clear tissue behavior or even specific gene-related clustering can be observed. This finding further supports that CU values show both mammal-and gene specific-related differences, which contribute to generate the CUB fingerprints. Indeed, it has been observed that extreme codon bias occurs in genes which underly specific functions, especially those functions related to processes which are unique in a given evolutionary lineage [4].

## CU values and CUB fingerprints in *HSA* DC and NDC genes

We grouped genes based on their propensity to be the site of pathogenic variations (mutations) causing rare diseases (DC or NDC genes). We profiled CU values in DC and NDC genes, preserving the tissue distinction (muscle, skin and kidney genes) across mammals. Fig 3, A to F panels, shows absolute CU values in DC and NDC muscle, skin and kidney genes, respectively, with no hierarchical clustering and identical codon type order. The CU values in these 6 panels show that the most frequent or rarest codon types are very similar in all genes and across mammals. We supported this observation in HSA by the Spearman correlation analysis, which demonstrated that the DC and NDC gene CU values correlate significantly in muscle, skin and kidney groups (p<0.05) (S1 Fig).

More CU value variability can be seen in codons with intermediate frequency, where some gene and tissue trends can be seen. In particular, muscle NDC genes do have higher CU values (a few yellow dots are visible, see Fig 3A and 3B), while muscle and kidney NDC and DC genes (Fig 3, panels A, B, E, F) have very similar CU values. To be noted, CAG is the most frequently used and UUA the less frequently used codon in all mammals.

Fig 3, panels G to L, show hierarchical clustering of CU values in the same gene categories as above. When looking at DC and NDC genes, recognizable CUB fingerprints can be observed. Muscle DC and NDC genes (Fig 3, panels G, H) show different fingerprints and clustering. Muscle DC genes have a compact cluster of extremely frequent codons (AAG, CAG, GAG) or extremely rare codons (UGG, UUA, CUA). CU values are homogeneous across mammals with clearly defined codon groups in terms of tree distance. In muscle NDC

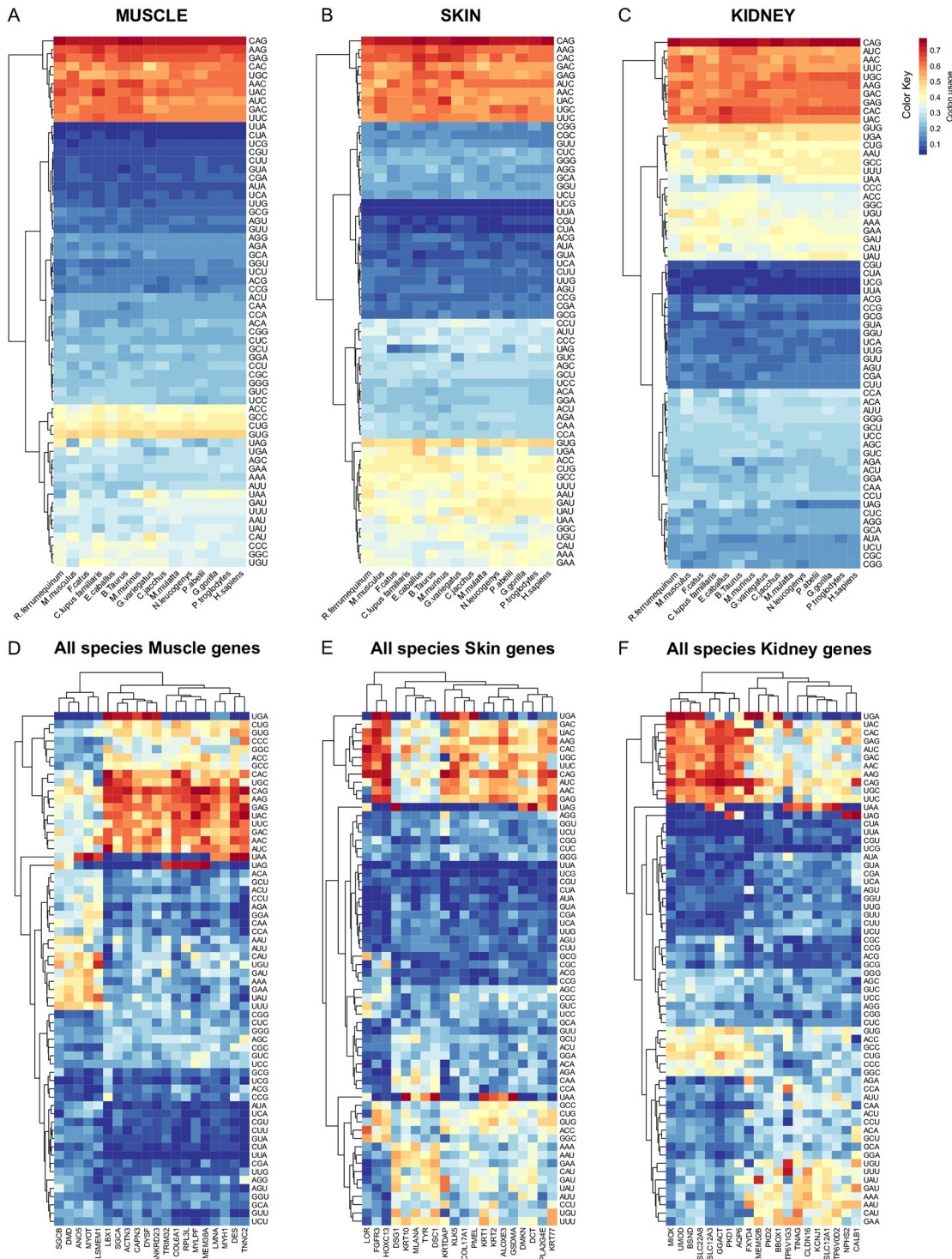

**Fig 2. CU values across mammals.** Heat plots were generated using R package gplots. Rows were clustered based on Euclidean distance. The color key of CU values varies from blue to red with low to high values of the CUB respectively. CU values in all 60 tissue-specific genes across 15 mammalian species in the metazoan phylogenetic tree are shown in panels A, B, and C, and CU values of single genes in all species are shown in panels D, E, and F. Tissue-specific CUB fingerprints are very evident with a conserved trend across mammals (panels A, B, C). Codons CAG, AAG, CAC, GAC, GAG, AUC, AAC, UAC, UGC, and UUC are the most frequently used codons (red spots) in all tissues and across mammals, while codons UUA, CUA, UCG, CGU, CUU, GUA, CGA, AUA, UCA, UUG, and GCG are the rarest codons (blue spots) in muscle and skin but not in kidney genes. Muscle (panel A) and skin (panel B) CUBs do have a similar fingerprint, although muscle gens have much more codons with lower CU values

compared to skin genes; kidney genes show a very different clustering. Panels D, E, and F show gene CU values grouping all mammals. Although clear clustering is poorly visible, tissues have different CUB fingerprints characterized by different gene-related dendrograms. Indeed, 13/30 genes in muscle (panel D), 4/40 genes in skin (panel E) and 8/20 genes in kidney (panel F) show high CU values clusters. Codon types also vary accordingly to CU values since high CU values are clustered in the above listed gene groups. This finding supports that CU values have a tissue-related fingerprint which is still maintained grouping all mammals, and that some gene-specific CU values and codon types are observable.

genes, the CUB fingerprint changes. Dark blue and red spots dominate, with a few codons with intermediate (yellow spots) CU values. This fingerprint indicates that a stronger CUB occurred in muscle NDC genes. Accordingly, the codon-based dendrogram in NDC, but not in DC, genes shows that higher and lower CU values are clustered together, underlining a possible different behavior across mammals.

Skin DC and NDC genes (Fig 3I and 3J) show similar CUB fingerprints with minor differences. Skin NDC genes show more codons with lower CU values (upper Fig 3J) and conversely less codons with higher or intermediate CU values (lower Fig 3J), compared to DC genes. This implies that CU intermediate values occur more frequently, an opposite finding of what seen in muscle genes. Indeed, skin DC genes fingerprint is similar to the one seen in muscle DC genes. We may conclude that both muscle and skin DC genes show a typical "no extreme CUB" pattern.

Kidney DC and NDC gene (Fig 3K and 3L) fingerprints and codon dendrograms greatly differ from the other two tissues, since the hierarchical distances between value clusters are opposite. Although conservation of CU values across mammals also occurs, the dendrogram's hierarchy of kidney genes shows a common ancestor for intermediate and low CU values, unlike muscle and skin genes, where two distinct lineages (high and low CU values) are visible, as already observed before (Fig 1C). Kidney DC and NDC gene CUB fingerprints are similar, although it has to be noted that NDC genes show a higher number of low CU values (yellow dots).

We also compared CU values among high-, medium- and low-expressed HSA genes. DC and NDC genes were therefore divided in three categories depending on gene expression level (for RNA fold and protein cut-off values, see S2 Fig). CUB fingerprints have high similarity, meaning that indeed, grouping genes for their expression level gets a similar CU values trend. We also analyzed whether there might be codons used more depending on gene expression level. Codons AAC, GAC, UGC, UAC, CAC, UUC, AUC, AAG, GAG and CAG are more frequently used both in high- and medium-expressed genes, while GUG is present only in highly expressed genes. Codons GAC, CAC, UUC, CAG, UAC, UGC, AAG, AUC, GAG, AAC, ACC, GGC, GUC, UCC and GCG are more frequently used in low-level expressed genes. Notably, while the majority of frequently used codons are similarly represented in highly, medium and low expressed genes, therefore with a CU not influenced by expression level, only a few codons with lower CU values such as UCC (Ser), ACC (Thr), GGC (Gly), GUC (Val) and GCG (Ala) are present in low-expressed genes. Similarly, codons with low CU values are not the same in high-expressed versus medium-expressed genes. Some highly expressed DC genes have more codons with higher CU values, like *DYS*, *LMNA* and *DES* (in skeletal muscle), *UMOD* and *PKD1* (in kidney) and *FGFR3* (in skin). The trend is opposite in medium-expressed genes, where some NDC genes, like *MLPF*, *TNNC2*, *TMEM3BA* (in skeletal muscle), and *NCLZ2*, *MCX* (in kidney), show higher CU values.

Interestingly, UAA, which is known to induce translation termination with higher speed and accuracy at the ribosomal level and can be read by both release factors eRF1 and eRF2, is

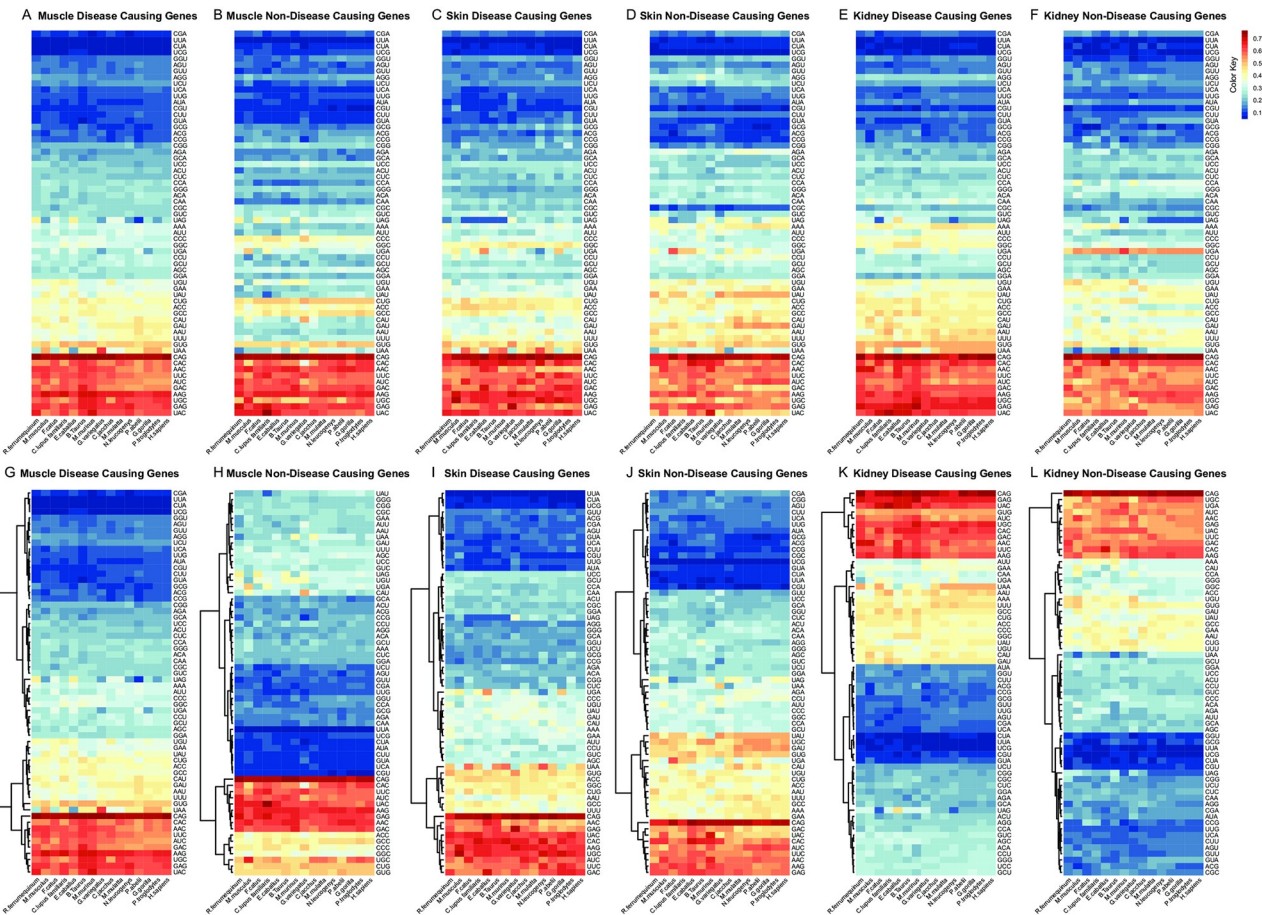

**Fig 3. Comparison of relative synonymous codon usage (CUB) values in disease-causing (DC) and non-disease-causing (NDC) genes across species.** Heat plots were generated using R package gplots. Rows were clustered based on Euclidean distance. The key color coding varies from blue to red with low to high values of the CUB respectively. We grouped genes based on their propensity to be the site of pathogenic variations (mutations) causing rare diseases (disease causing or DC genes). Panels A to F shows all types of codons with no hierarchical clustering, listed in the same order in all genes and tissues, and sub-grouped in DC and NDC. CU values in these panels show that the most frequent or rarest codon types are overlapping in gene groups. Two groups of dark blue and dark red color key codons occur in all 6 panels, indicating that absolute frequency of codon types is similar also across genes and mammals, possibly supporting an evolutionary trend of CUB. Nevertheless, CU value higher variability can be seen in codons with intermediate frequency (light blue to yellow key colors) which differ across gene groups. Muscle NDC genes (panel B) do have the lower number of intermediate (yellow key color) CU values, followed by skin DC (panel C) and kidney NDC (panel F). CAG is the most frequently used and UUA is the less frequently used codon in all genes and across mammals. Panels G-L show hierarchical clustering of CU values in the same categories and groups. Clearly recognizable CUB fingerprints can be observed in DC and NDC genes. This is massively evident in muscle genes (panel G and H) and also partially in kidney genes (panel K and L). DC muscle genes have compact clusters of extremely frequent codons (AAG, CAG, GAG) and extremely rare codons (UGG, UUA, CUA), a trend conserved across mammals and with clearly defined codon groups in terms of tree distance. Dark blue and red (more rare and frequent codons) colors predominate in muscle NDC genes, with a few codons with intermediate (yellow color key) values. This suggests that a strong CUB has occurred in NDC genes. Accordingly, muscle NDC genes show higher and lower CU values clustered together, suggesting a possible different evolutionary trend. Skin DC and NDC genes (panels I-J) show similar CUB fingerprints with a few differences. Skin NDC genes show overall lower CU values (panel J, upper side) and conversely more codons with intermediate CU values (panel J, downside) compared to DC genes. Kidney DC and NDC gene (panels K-L) CUB fingerprints greatly differ from the other two groups since the hierarchical clusters are opposite. Although conservation of CU values across mammals also occurs, the hierarchy of kidney gene dendrograms shows a common ancestor for intermediate and low CU values and not two distinct lineages (high and low CU values) as observable in muscle and skin genes. The two kidney CUB fingerprints are somehow similar; however, DC genes show a higher number of very low CU values.

the most used stop codon in highly expressed genes [36, 37]. UAG and UGA have a similar frequency in all genes.

## Prioritization of the most differently used codons among tissue-specific genes

We applied the same strategy of using CU values calculation to compare different human genes to identify eventual codons which may be more differentially/preferentially used in certain disease-causing genes, compared to non-disease-causing genes. If confirmed, this may imply that some synonymous codons may be preferred in the coding sequence of human genes which mutations do cause rare diseases. This finding may represent an additional criterion to be added to in silico tools or it may be also become a novel metric to design algorithm useful for synonymous variants interpretation.

Based on the codon usage p-value ($P < 0,05$), we prioritized codons with very different CU values among gene tissues (S11 and S12 Tables). Results are shown in Fig 4. Five codons were prioritized, being used significantly differently in the three human tissues: CGU (Arg), CCA (Pro), GAC (Asp), GAU (Asp) and GUA (Val) (Fig 4A–4E). CCA and CGU are the least frequently used codons in muscle, GUA in kidney, and GAU and GAC are the most frequently used codons in skin. Considering DC and NDC genes, further differences in CU values can be observed for CGU and CCA, which are more frequently used in DC genes, and for GUA and GAU, which, though less significantly, are more frequently used in NDC genes (Fig 4F–4L). These data suggest that the CU values might be influenced by phenotype and gene propensity to cause genetic diseases.

The CU value trend of these 5 codons seems to be conserved across mammals (Fig 4F–4L).

Finally, we counted the number of extremely biased codons in genes and across mammals. We found that DC genes retain a higher multiple codon usage, with only a few extremely biased codons, compared to NDC genes (Table 1, and Fig 5A and 5B).

## DMD unique CUB behavior

We used the calculated CU values and we mapped 2828 known pathogenic missense and nonsense mutations in the dystrophin (*DMD*) gene, taken from either public (LOVD) [38] or internal databases [21], onto the *DMD* codons. We called our approach mutations' "mapping-on-codon".

We verified whether rarely/frequently used DMD codons are consistently rarely/frequently site of proven pathogenic missense and nonsense variations. This finding may suggest that some codons might be more prone/less prone to be site of pathogenic variations in a gene specific (*DMD*, in this case) context, and perhaps the "more-less mutated codons" might be relevant for the (*DMD*) gene translation capacity, and/or for other translation-related functions, meaning that they should be considered when artificial gene codon optimization is carried out.

The *DMD* gene does not have an extreme CUB and maintains all codon types used in its coding sequence, also across the mammals studied. We counted the number of codon type in *DMD* coding sequence and identified only 4 extremely biased codons, UCG, CCG, ACG and GCG, accordingly to the known bias cut-off parameter, based on codon redundancy of 2, 3, 4 and 6 triplets [39], (Fig 6A).

Fig 6B shows the number of mutations occurring at all codon types. Each number, on the top of the bars, represents how many times that specific codon was the site of a *DMD* missense or nonsense variation. Interestingly, these numbers are not related to CU values. Although CCG (Pro), UCG (Ser), GCG (Ala) and ACG (Thr) in red in Fig 6A and 6B, are, expectedly,

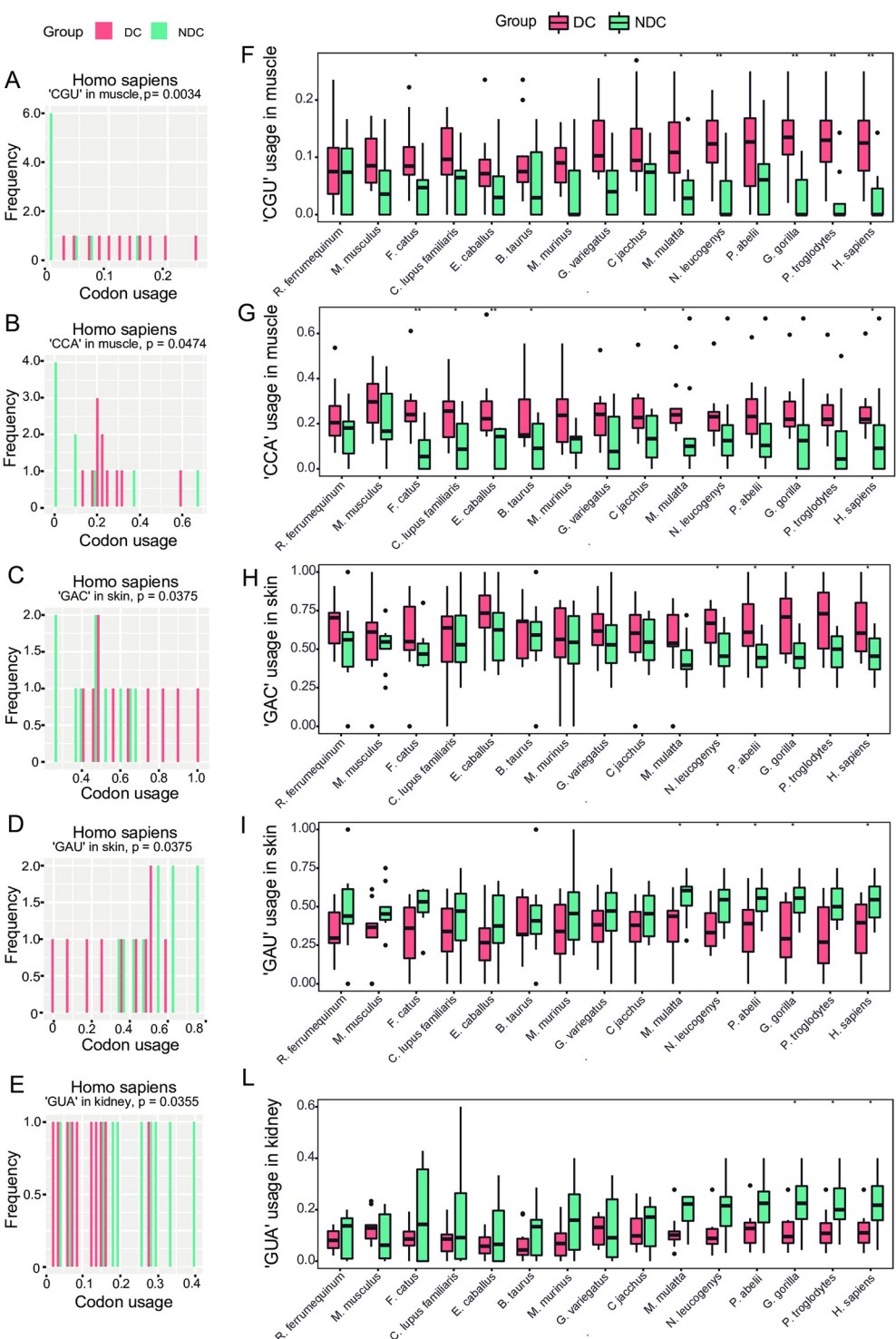

**Fig 4. The 5 extremely biased codons, differentially used by *Homo sapiens* in DC vs NDC genes, and among tissues.** Left graphs (A, B, C, D, E) show the codon frequency on the x-axis and the number of genes that use that specific codon (with the frequency annotate in x-axis) on the y-axis. Right graphs (F, G, H, I, L) show CU values in all studied mammals on the x-axis and codon usage values in gene tissues on the y-axis. Five codons were the most differentially used in *HSA*, CGU (Arg), CCA (Pro), GAC (Asp), GAU (Asp), and GUA (Val). Pink bars indicate DC genes, blue bars indicate NDC genes. CGU (A, F) is the least frequently used codon in NDC muscle genes; CCA (B, G) is the least frequently used codon in DC muscle genes; GAC (C, H) is the most frequently used codon in DC skin genes; GAU (D, I) is the most frequently used codon in NDC skin genes; GUA (E, L) is the least frequently used codon

in DC kidney genes. Comparing CU values of these 5 extremely biased codons between DC and NDC gene groups and across mammals, the trend toward a heavy codon extremization during evolution can be appreciated. During evolution, CGU and CCA became more used in DC muscle genes, GAC became more used in skin DC genes, while GAU became more used in skin NDC genes, and GUA became more used in NDC kidney genes. This suggests a codon type-specific, disease-driven CUB, apparently conserved across mammals.

less frequently site of mutations, being the rarest codons used by the *DMD* gene, other codon CU values do not correlate with mutation occurrence. This is the case of CAG (Glu), which shows intermediate CU values but is the codon more frequently site of *DMD* mutations, and UAU (Tyr) and UUU (Phe), which are the most frequently used DMD codons (Fig 6A) but are rarely site of mutations (Fig 6B).

## Discussion

We calculated CU values in three small groups of genes which are tissue-specific and mostly expressed in skin, kidney and skeletal muscle. We then innovatively compared CU values between DC and NDC genes, and across mammals, therefore using a disease-driven approach, to explore CU values and CUB behavior.

We confirmed that tissue-specificity influences CUB, and by calculating CU values in high-, medium- and low-expressed genes we observed a tissue-specific trend, in keeping with previous reports [14; 40]. Interestingly, some codons are more represented in high- or low-expressed genes, a fact possibly related to a positive selection of codons with higher/lower translational capacity during evolution, and which depends on gene role in specific tissues and/or organs. Consistently with this hypothesis, highly expressed DC genes show higher CU values, finding that might suggest a key role of some codons in tissue specific DC genes, by conferring their more efficient translation rate in some tissues. Conversely, and supporting a key role of CU, the UAA stop codon is preferentially used in muscle genes and, generally, in highly expressed genes, probably reflecting the need of an optimal codon ribosomal recognition to efficiently stop translation [41]. We therefore confirmed that, by calculating CU values and comparing them in small groups of selected, highly tissue-specific, human genes, CU values' differences can be observed, and are associated with tissue and gene expression level.

### Tissue-specific fingerprints in HSA genes

Comparing CU values by hierarchical clusters in muscle, skin, and kidney *HSA* genes, different patterns can be seen. Rarely or frequently used codon types vary among the three tissues, muscle and skin genes being more similar in terms of clusters and hierarchy of CU values. These different CUB fingerprints may be due to the tissue specificity of the analyzed genes, and we may speculate that those genes having tissue-related functions which may require higher or lower translation efficiency, use different synonymous codons. Interestingly, many muscle and kidney genes share the same clustered, frequently used, codons, supporting that these might be key codons to regulate tissue-specific translation. Notably, muscle and kidney, together with liver and lung, are parenchymal tissues. These tissue types undergo a similar poor organ regeneration ability, because of evolutionary trade-offs, which are especially related to balancing effects between the immune system and the shape of physiological and pathological processes [42]. Additionally, a similar high CU value clustering is even more evident for some muscle (COL6A1, RPL3L, MYLPF, TMEM38A, TNNC2, LMNA, DES, LBX1, SGCA, ANKRD23, CAPN3, DYSF, ACTN3) and kidney (UMOD, BSND, SLC22A8, MIOX, AQP6, PKD1, SLC12A3, GGACT) genes, underlining some gene-specific and/or organ related functions.

**Table 1. Number of extremely biased codons ("zero-codons") found in DC and NDC tissue-specific genes across mammals.** Table 1 shows the number of extremely biased codons ("zero-codons") in gene tissues and across mammals. Genes are grouped in DC and NDC. The number of "zero-codons" is based on codons usage (CU) value that we calculated in each group of genes and across mammals. The DC genes retain a multiple codon usage (low CUB), therefore with a few "zero-codons" compared to NDC genes. Notably, the number of "zero-codons" is much higher in muscle and kidney NDC and skin DC genes, therefore with a disease-driven behavior. CUB increased progressively during evolution, although not at the same extent in all gene groups. The most remarkable difference in the "zero-codon" number is between *HSA* kidney DC (11) and NDC (94) genes and muscle DC (49) and NDC (109) genes.

| | Rhinolophus_ ferrumequinum | Mus_ musculus | Felis_ catus | Canis_ lupus | Equus_ caballus | Bos_ taurus | Microcebus_ murinus | Galeopterus_ variegatus | Callithrix_ jacchus | Macaca_ mulatta | Nomascus_ leucogenys | Pongo_ abelii | Gorilla_ gorilla | Pan_ troglodytes | Homo_ sapiens |
|---|---|---|---|---|---|---|---|---|---|---|---|---|---|---|---|
| **NDC** | | | | | | | | | | | | | | | |
| kidney | 95 | 93 | 97 | 98 | 111 | 108 | 100 | 83 | 105 | 106 | 116 | 120 | 118 | 117 | 114 |
| muscle | 85 | 86 | 91 | 97 | 105 | 99 | 100 | 89 | 89 | 96 | 97 | 107 | 97 | 98 | 104 |
| skin | 62 | 62 | 54 | 62 | 61 | 75 | 64 | 50 | 57 | 75 | 65 | 64 | 59 | 62 | 63 |
| **DC** | | | | | | | | | | | | | | | |
| kidney | 39 | 35 | 33 | 28 | 42 | 38 | 41 | 45 | 36 | 36 | 36 | 32 | 31 | 31 | 31 |
| muscle | 46 | 35 | 46 | 53 | 49 | 47 | 52 | 46 | 46 | 45 | 44 | 51 | 44 | 45 | 45 |
| skin | 37 | 66 | 73 | 84 | 67 | 37 | 74 | 52 | 65 | 74 | 58 | 74 | 64 | 57 | 78 |

Panel A

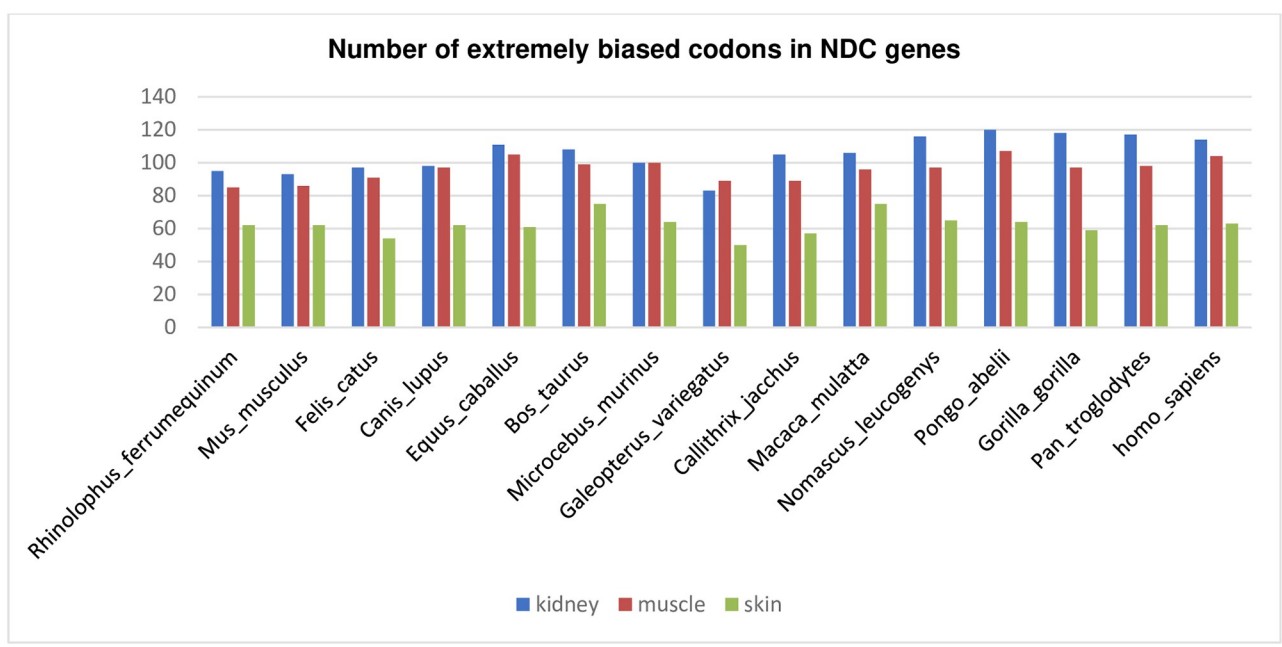

Panel B

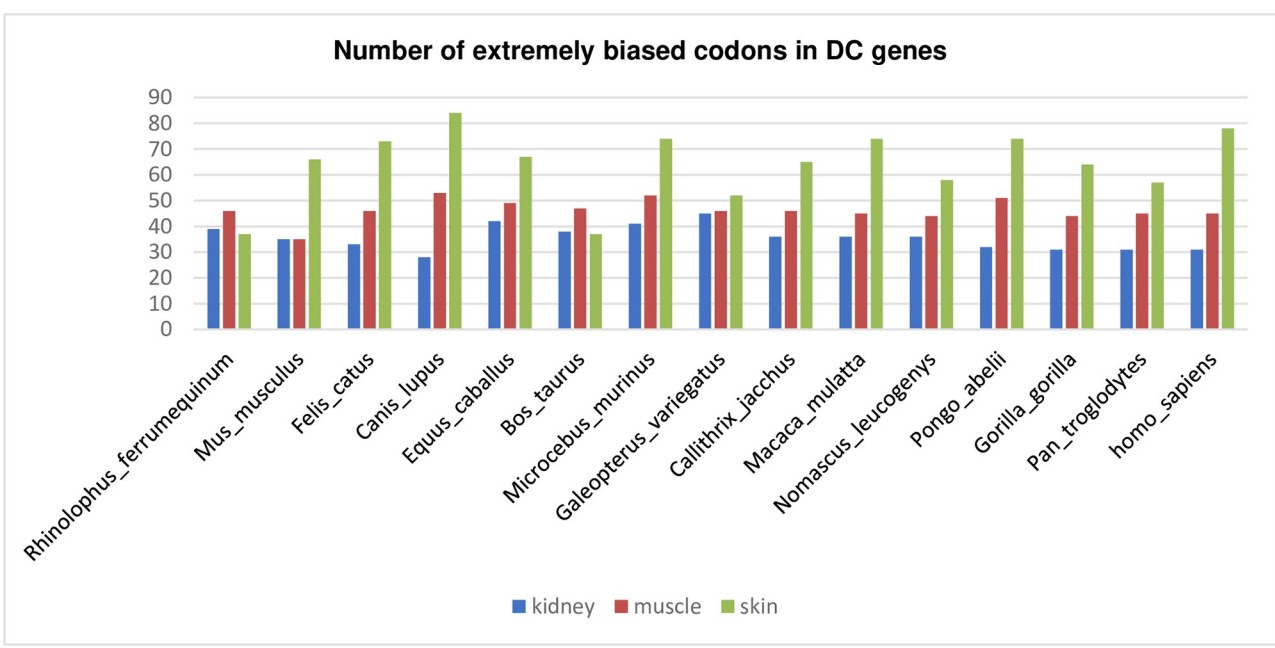

**Fig 5. Number of extremely biased codons ("zero-codons") found in DC and NDC tissue-specific genes across mammals.** Panel A and B show on graphs the same data reported in Table 1, to better appreciate the CUB trend and the number of "zero codons". Specifically, the number of extremely biased codons ("zero-codons") in NDC and DC gene tissues are shown in panel A and panel B, respectively. On the X-axis are listed the mammals' species in an evolutionary order; in the Y-axis the cardinal number of "zero codons". Blue, red, and green bars represent kidney, muscle, and skin tissues genes, respectively.

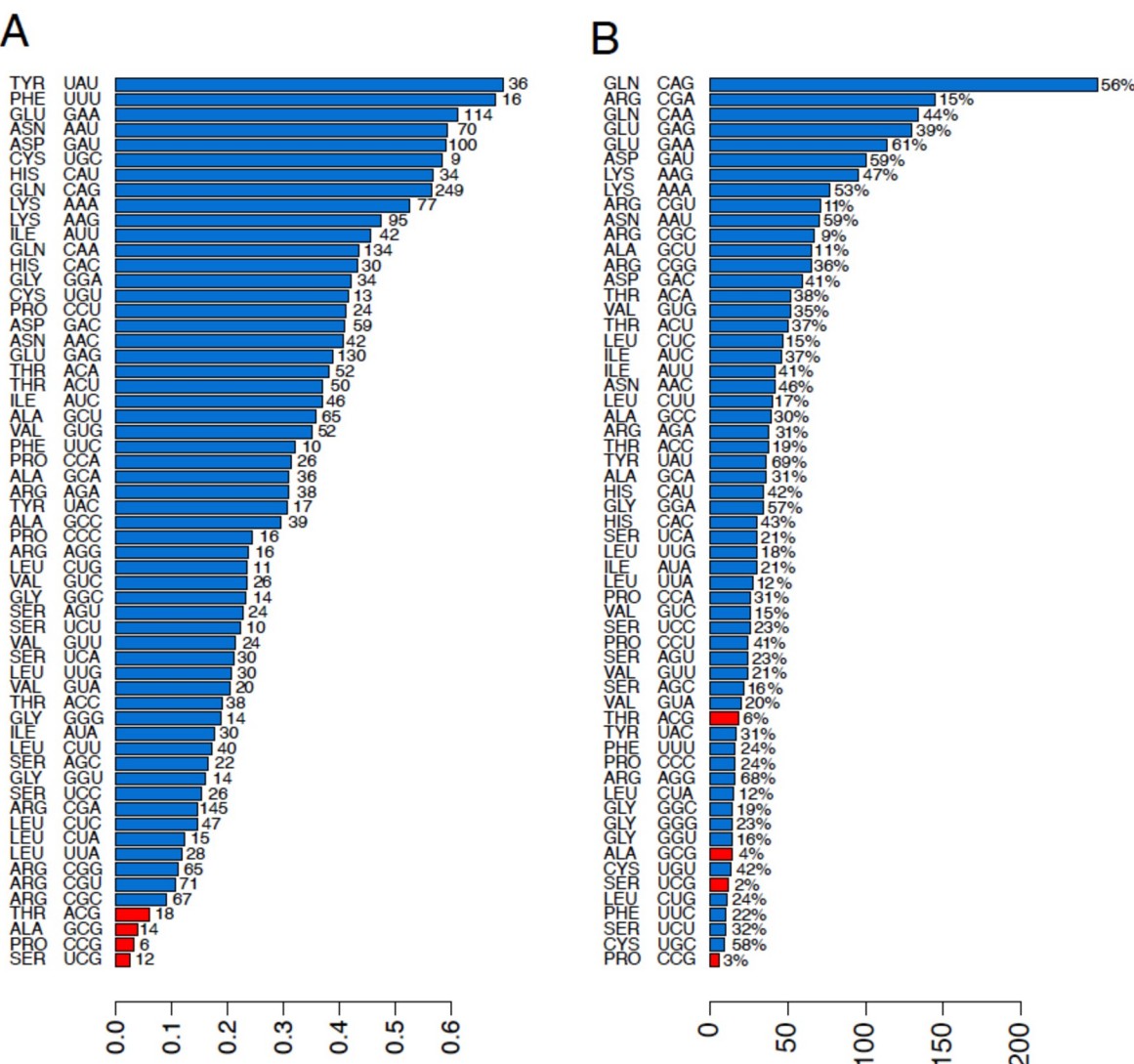

**Fig 6. Human DMD gene codon usage and mutations "mapping-on-codon" approach.** Panel A Human DMD codon usage values and mutation percentages. The bars represent the CU values in *DMD* gene in *HSA*. On the x-axis, codon types and the relative amino acid are listed, on the y-axis CU values are reported. Red bars represent the 4 *DMD* rarely used codons UCG, CCG, ACG and GCG, according to our cut-off, which is based on codon redundancy of 2, 3, 4 and 6 triplets (see Methods). On the top of bars (right side), the numbers of missense and nonsense mutations occurred at the relative *DMD* codons are reported. All codons are still used by the *DMD* human gene, and the frequency of mutation occurrence is not related to the codon usage values. Examples are UAU, which is the most used codon, but with only 36 "mapped" mutations, or CGA (Arg) which is rarely used but has 114 "mapped" mutations. Panel B. Mapping of human DMD missense and nonsense mutations on codon types. On the x-axis, there are codons and the relative amino acid, on the y-axis, there is the number of mutations which have occurred and "mapped". Red bars represent the 4 rarely used codons in the DMD human gene according to our cut-off. On the top of the bars there is the percentage of CU values. The number of occurring mutations and the CU values are not strictly related. Examples are CAG, which is the "more frequently" mutated codon (56%), with high CU values, and UGC (Cys), rarely site of mutations, but with very high CU value.

These genes, interestingly, are all DC genes, fact that is consistent with our DC and NDC genes comparison results (see below). Accordingly, most skin genes show low or intermediate CU values, with the few higher CU values never clustered. Skin is known to be an "immunological microenvironment" which regulates cell regeneration, thus its opposite CUB trend, compared to muscle and kidney genes, might reflect its different organ and developmental function [43].

## CUB fingerprints across mammals

CUB fingerprint differences we have observed in muscle, kidney and skin genes are also visible across mammals. CUB evolutionary conservation has been widely studied in *HSA*, but not in such a genetic granularity. Muscle and skin, but not kidney, genes show similar CUB patterns with two major hierarchy lineages, one for more frequent and one for rarest codons. These trends suggest that CUB in muscle and skin genes may have possibly followed some common evolutionary pathways. Muscle genes are extremely important in mammals, where they contribute to about 80% of body mass. In *HSA*, the acquisition of bipedalism has certainly required a robust selective force to drive the muscle reshaping, especially for muscles related to limb girdles [44]. Common origin between skin and striated muscle can be found in the panniculus carnosus, a thin striated muscular layer attached to the skin and fascia of most mammals, which provides support for twitching and contraction muscle functions. Panniculus carnosus is still conserved in humans, though it is considered of no functional significance, and it is a remnant of evolution, reflecting the common origin of muscle and skin [45]. Supporting this link, a cohort of rare musculocutaneous syndromes due to mutations in RAS/MAPK pathway genes were described in humans and since of their similar CUB fingerprints, other muscle and skin DC genes would deserve to be studied by our strategy [46].

## CUB fingerprints in disease-causing genes

By comparing CU values in rare disease-causing genes, we showed that, although tissue-specific pathways are still well recognizable, some DC genes (especially in muscle) have different CUB fingerprints compared to NDC genes.

In general, DC genes show less extreme CU values compared to NDC genes, fact very evident if we calculated CU values without hierarchical clustering. Muscle and, less evidently, skin and kidney DC genes do show more intermediate values compared to NDC genes, suggesting a different CUB behavior.

Above all, muscle DC genes show the most recognizable disease-related fingerprint, suggesting that a "disease-driven" CUB may partly prevail on the tissue specificity CUB choices in muscle. This is a novel observation which would deserve more analysis to understand the role of codon choices in human disease genes.

Skin and kidney DC and NDC genes show less evident differences. Nevertheless, some codons, such as AAG, CAG, and GAG, have higher CU values in DC genes, raising the possibility that their frequency might be perhaps disease-driven.

It is known, although reasons not fully understood, that CUB increases during evolution and becomes extreme with the complete lack of some codon representation, codons we defined "zero codons" [47]. The disease-driven muscle DC genes CU fingerprint we observed suggests a different natural selection pressure, as already identified in some human gene categories [48].

A gene-specific CUB was identified in a few human disease genes, suggesting that some synonymous variations may have functional implications. In *CFTR* and *GATA4* genes (which mutations cause Cystic Fibrosis and a congenital heart disease, respectively), synonymous mutations may alter translational kinetics and protein folding by introducing non-optimal or rare codons [49, 50]; high ACT, AGG, ATT and AGC, or AGA CU values were seen in *HPRT1* (which mutations occur in Lesch-Nyhan Syndrome) and *GALC* (which mutations cause Krabbe disease) genes, [51, 52]; finally *BRCA1* and *BRCA2* (major genes involved in Mendelian breast cancers) genes show an extremely low CUB compared to other oncogenes [53]. Taking together these and our data, we may hypothesize that some DC genes may have undergone a different CUB pressure during evolution. In support of this evidence, it is known that

some disease genes, such as the *DMD* gene [54], still have a high mutation rate, which might impact on codon type frequency. Notably, it has been very recently shown that gene/protein evolution might occur not only in terms of driving evolutionary forces but also in relationship with the diseases their mutations may cause [55].

Therefore, in our studied DC genes we showed that CUB might be disease-driven, suggesting that comparing CU values in these genes, which are more than 6000 already identified so fare in humans, might be a good strategy to understand codon usage governing rules and their relationship with diseases.

## Codon prioritization and DMD gene CUB behavior

Five codons, GUA (Val), GAU (Asp), GAC (Asp), CCA (Pro), and CGU (Arg) showed the most different CU values among *HSA* genes and among tissues.

GAC and GAU are the top used codons in skin genes, contrariwise CAA and CGU and GUA are the least used codons in muscle and kidney genes, respectively. Interestingly, these three last codons, together with codon ATC (Ile), have a significant positive correlation with gene expression due to their high GC content [56]. This finding supports the tissue-oriented over-usage of some codons, like the one we found for these 5 codons in our study, likely reflecting a specific selection process.

Supporting our hypothesis about disease-driven CUB fingerprints, the tissue-specific usage of these 5 top codons is also disease-driven. CCA and CGU are the most used muscle codons with higher CU values in muscle DC genes. Similarly, GAC is a most-used skin DC genes codon, while GAU codon is more frequently used in NDC genes. This disease-driven trend seems to be also conserved across mammals.

A few available reports suggest that specific codon overrepresentation may characterize Mendelian disorder genes [57, 58] and, according to the directional mutation pressure theory, might be due to a negative selective pressure at codons conferring higher risk of mutation rates occur [59]. It is also known that some *HSA* genes show stronger or lower CUB depending on their function [41]. We therefore hypothesize that some DC genes with unique functions may have had a different evolutionary transition in terms of CUB, for the reasons above mentioned.

The *DMD* gene [60] is the only *HSA* gene which does not have any "zero codons" and still uses all codons in its coding sequence. This trend is conserved across mammals. DMD mutations, including nonsense and missense variations, cause Duchenne muscular dystrophy (DMD, OMIM* 300377) a rare, severe and fatal muscle dystrophy, or the milder form, Becker muscular dystrophy (BMD; OMIM *300376), both inherited as X-linked recessive diseases, with an incidence of 1:5000 newborn males.

By our "mapping-on-codons" approach of the 2828 *DMD* pathogenic missense and nonsense variations causing DMD or BMD, we found patchy correlations between variation frequency at codons and CU values. The least frequently used *DMD* codons (UCG, CCG, ACG and GCG) are GC-rich and indeed rarely host mutations, as expected [14]. However, other GC-rich codons like CGC, are rarely used in *DMD* but are frequently site of mutations (67) and, conversely, the very frequently used codon UGC only hosts 9 mutations. CAG is the most frequently DMD mutated codon (56,5% with 249 mutations) but not the most used. Therefore, rare codons might be often site of mutations, and vice versa. Interestingly, the *DMD* locus architecture provides possible explanations of the unique lack of "zero codon" behavior. It is known that length of introns and proteins, the expression patterns and the CUB values are genomic variables that influence the gene evolutionary rate [59]. Genes with low mRNA/protein expression levels tend to evolve rapidly, have large introns, code for heavier proteins, and

have very low CUB. The *DMD* gene meets all these rules: it has huge introns, encodes a high molecular weight protein, which is highly tissue-specific but poorly abundant and consistently shows an absence of extreme CUB and no "zero codons". Based on these metrics, the *DMD* gene may have rapidly evolved during evolution, and we may infer that a similar behavior has occurred for other DC genes which show similar low-CUB characteristics like those observed in many kidney and skin DC genes. Further supporting our theory, eukaryotes show a negative correlation between gene length and CUB, while in mosquitoes, an antagonistic relationship between CUB and the intron number and length was described and, more intriguingly, intron-less genes have a very high CUB with high "zero codon" level [5, 61].

Our mapping-on-codon approach may also be useful in interpreting the functional meaning of the many synonymous changes that occur in the DMD, and in other DC, genes, and which remain of uncertain or unknown significance. In this study, we have analyzed only pathogenic variations by our "mapping-on-codon" approach, but synonymous changes can be studied as well, to evaluate whether a relationship exists between these variations and CU values of their codons. Indeed, a CUB-oriented, tissue-oriented, or even disease-driven, algorithm to define the synonymous change meaning might be attractive to be developed.

As a final reflection, more understanding about CUB fingerprints in DC genes may have implications in codon optimization methods. Indeed, DC genes, especially in muscle, show a low CUB. This means that muscle DC genes still apply the full codon redundancy and, consequently, that the corresponding synthetic genes designed for gene therapies will need a disrupting codon optimization via Codon Adaptation Index (CAI) appliance [62, 63]. Oppositely, NDC genes show spontaneous high CUB, which leads to many "zero-codons" in the coding sequences, with poor codon redundancy. Since we showed that DC genes (especially muscle) have a disease-driven CUB, true applicability to "all" disease genes of the currently used computation algorithms for codon optimization and gene therapy approaches might be questioned [64]. Indeed, some rare, non-optimal, codons may have to be preserved for protein translation efficiency or, more interestingly, for gene- and tissue-specific expression regulation as reported, although not in humans [19, 65], and CAI may benefit of using other novel metrics.

## Conclusions

In our study, we presented a novel strategy to compare CU values in gene categories. We found that CU values hierarchical clusters vary depending on tissue specificity but also on gene-related phenotype, with a disease-driven trend, which seems to be conserved among the mammals studied. We think that our preliminary findings are encouraging and suggest that calculating and comparing CU values in many human disease genes might be valuable to unravel novel codon usage characteristics. This knowledge may impact on synonymous variation interpretation and synonymous codon translation capacity when designing algorithms for synthetic gene development.

## Supporting information

**S1 Fig. Spearman correlation analysis between DC and NDC genes in *HSA* muscle, skin and kidney tissues' genes.** The test demonstrated that DC and NDC genes CU values correlate significantly in muscle, skin and kidney (p<0.05).
(PDF)

**S2 Fig. Heat plots were generated using R package gplots.** Rows were clustered based on Euclidean distance. The color coding varies from dark blue to red with low to high CU values respectively. CUB fingerprint and CU values among high-, medium- and low-expressed HSA

genes. DC and NDC genes were considered depending on their expression level. CUB finger-prints have high similarity, meaning that indeed, grouping genes for their expression level yields a similar CU value trend. Codons AAC, GAC, UGC, UAC, CAC, UUC, AUC, AAG, GAG and CAG are more frequently used both in high- and medium-expressed genes while GUG is only present in highly expressed genes. Codons GAC, CAC, UUC, CAG, UAC, UGC, AAG, AUC, GAG, AAC, ACC, GGC, GUC, UCC and GCG are more frequently used in low-level expressed genes. A few codons have lower CU values such as UCC (Ser), ACC (Thr), GGC (Gly), GUC (Val) and GCG (Ala) in low-expressed genes. Some highly expressed DC genes have more codons with higher CU values, like *DYS*, *LMNA* and *DES* (muscle), *UMOD* and *PKD1* (kidney) and *FGFR3* (skin). In medium-expressed genes, the trend is opposite, with some NDC genes that show higher CU values like *MLPF*, *TNNC2*, *TMEM3BA* (muscle) and *NCLZ2*, *MCX* (kidney). Interestingly, UAA is the most used stop codon in highly expressed genes since it induces translation termination with higher speed and accuracy at the ribosomal level and can be read by both release factors eRF1 and eRF2 [31, 32]. UAG and UGA have a similar frequency in all tissue genes and expression levels.
(PDF)

**S1 Table. The table shows the list of considered species named in both the scientific and common name (on the left) and their link in the NCBI browser (on the right).**
(DOCX)

**S2 Table. The table shows the list of *Homo sapiens* genes, prioritized in skeletal muscle tissue, using the Human Protein Atlas database (https://www.proteinatlas.org/).** In order to prioritize muscle genes, we selected those with higher expressions from the skeletal muscle-enriched genes list of the Human Protein Atlas database (https://www.proteinatlas.org/search/tissue_specificity_rna:skeletal%20muscle;Tissue%20enriched+AND+sort_by:tissue+specific+score+AND+show_columns:groupenriched). All data (RNA, TS, TPM, and Protein expression scores) were also obtained by the Human Protein Atlas database. *RNA TS TPM indicates RNA level reported as mean TPM (transcripts per million), in referred tissue, skeletal muscle in this case. **Protein expression scores are based on a best estimate of the "true" protein expression from a knowledge-based annotation in the selected tissue, skeletal muscle in this case. ***Tissue specificity is based on data found in the graph called "HPA tissue dataset", a sub-category of the "RNA sample summary" section in the HPA site, for each gene. The RNA summary section shows normal distribution of individual samples across the datasets of multiple RNA-seq analyses visualized with box plots. "Only" is used for a gene transcript present only in the specific tissue (skeletal muscle). "Predominantly" is used when the majority of a gene transcript is present in the specific tissue (skeletal muscle). "All" is used for a gene transcript present in all tissues.
(DOCX)

**S3 Table. The table shows the list of *Homo sapiens* genes, prioritized in skin tissue, using the Human Protein Atlas database (https://www.proteinatlas.org/).** In order to prioritize skin genes, we selected those with higher expressions from the skin-enriched genes list of the Human Protein Atlas database (https://www.proteinatlas.org/search/tissue_specificity_rna:skin;Tissue%20enriched+AND+sort_by:tissue+specific+score+AND+show_columns:groupenriched). All data (RNA, TS, TPM, Protein expression scores and Tissue specificity) were also obtained by the Human Protein Atlas database. *RNA TS TPM indicates RNA level reported as mean TPM (transcripts per million), in referred tissue, skin in this case. **Protein expression scores are based on a best estimate of the "true" protein expression from a knowl-edge-based annotation in the selected tissue, skin in this case. ***Tissue specificity is based on

data found in the graph called "HPA tissue dataset", a sub-category of the "RNA sample summary" section in the HPA site, for each gene. The RNA summary section shows normal distribution of individual samples across the datasets of multiple RNA-seq analyses visualized with box plots. "Only" is used for a gene transcript present only in the specific tissue (skin). "Predominantly" is used when the majority of a gene transcript is present in the specific tissue (skin). "All" is used for a gene transcript present in all tissues.
(DOCX)

**S4 Table. The table show the list of *Homo sapiens* genes, prioritized in kidney tissue, using the Human Protein Atlas database (https://www.proteinatlas.org/).** In order to prioritize kidney genes, we selected those with higher expression from the kidney-enriched genes list of the Human Protein Atlas database (https://www.proteinatlas.org/search/tissue_specificity_rna:kidney;Tissue%20enriched+AND+sort_by:tissue+specific+score+AND+show_columns:groupenriched). All data (RNA, TS, TPM, Protein expression scores and Tissue specificity) were also obtained by the Human Protein Atlas database. *RNA TS TPM indicates RNA level reported as mean TPM (transcripts per million), in referred tissue, kidney in this case. **Protein expression scores are based on a best estimate of the "true" protein expression from a knowledge-based annotation in the selected tissue, kidney in this case. ***Tissue specificity is based on data found in the graph called "HPA tissue dataset", a sub-category of the "RNA sample summary" section in the HPA site, for each gene. The RNA summary section shows normal distribution of individual samples across the datasets of multiple RNA-seq analyses visualized with box plots. "Only" is used for a gene transcript present only in the specific tissue (kidney). "Predominantly" is used when the majority of a gene transcript is present in the specific tissue (kidney). "All" is used for a gene transcript present in all tissues.
(DOCX)

**S5 Table. The table reports the CUB and CU values for each amino acid in all the DC muscle genes considered.** Every gene is divided in a specific page (on the bottom) across species (on the top).
(XLSX)

**S6 Table. The table reports the CUB and CU values for each amino acid in all the NDC muscle genes considered.** Every gene is divided in a specific page (on the bottom) across species (on the top).
(XLSX)

**S7 Table. The table reports the CUB and CU values for each amino acid in all the DC skin genes considered.** Every gene is divided in a specific page (on the bottom) across species (on the top).
(XLS)

**S8 Table. The table reports the CUB and CU values for each amino acid in all the NDC skin genes considered.** Every gene is divided in a specific page (on the bottom) across species (on the top).
(XLS)

**S9 Table. The table reports the CUB and CU values for each amino acid in all the DC kidney genes considered.** Every gene is divided in a specific page (on the bottom) across species (on the top).
(XLS)

**S10 Table. The tables report the CUB and CU values for each amino acid in all the NDC kidney genes considered.** Every gene is divided in a specific page (on the bottom) across species (on the top).
(XLS)

**S11 Table. The table lists all the codon usage p-value, calculated in all the considered** *Homo sapiens* **genes, both DC and NDC genes in the three tissues (muscle, skin and kidney).**
(TXT)

**S12 Table. The table lists all the codon usage p-value, calculated all the considered genes across species, in the three tissues (muscle, skin and kidney).**
(XLSX)

## Acknowledgments

AF is Chair of the Genetic Task of the European Reference Network Euro-NMD (www.ern. euro-nmd.eu). We wish to acknowledge the work of the *DMD* gene database curator, Prof. Johan denDunnen (LOVD, www.lovd.nl). The strategy we describe here for CU values analysis and comparison is patented at UNIFE with the provisional number Rif. Sib. BI5666R.

## Author Contributions

**Conceptualization:** Rachele Rossi, Alessandra Ferlini.

**Data curation:** Rachele Rossi, Mingyan Fang, Lin Zhu, Alessandra Ferlini.

**Formal analysis:** Rachele Rossi, Mingyan Fang, Lin Zhu, Cristina Flesia, Alessandra Ferlini.

**Funding acquisition:** Chongyi Jiang, Cong Yu, Chao Nie, Wenyan Li.

**Investigation:** Rachele Rossi.

**Methodology:** Rachele Rossi, Mingyan Fang, Lin Zhu, Alessandra Ferlini.

**Software:** Mingyan Fang.

**Supervision:** Rachele Rossi, Chongyi Jiang, Cong Yu, Cristina Flesia, Chao Nie, Wenyan Li, Alessandra Ferlini.

**Validation:** Rachele Rossi, Mingyan Fang, Chongyi Jiang, Cong Yu, Chao Nie, Wenyan Li, Alessandra Ferlini.

**Visualization:** Alessandra Ferlini.

**Writing – original draft:** Rachele Rossi, Alessandra Ferlini.

**Writing – review & editing:** Mingyan Fang, Alessandra Ferlini.

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
