## [Decision Letter · Decision Letter 0]

6 Dec 2021

PONE-D-21-32048Calculating and comparing codon usage values in rare disease genes highlights codon clustering with disease-and tissue- specific hierarchyPLOS ONE

Dear Dr. Ferlini,

Thank you for submitting your manuscript to PLOS ONE. After careful consideration, we feel that it has merit but does not fully meet PLOS ONE’s publication criteria as it currently stands. Therefore, we invite you to submit a revised version of the manuscript that addresses the points raised during the review process.

Please address both reviewers comments.

We look forward to receiving your revised manuscript.

Kind regards,

Yanbin Yin

Academic Editor

PLOS ONE

Journal Requirements:

[The SOLVE-RD (grant agreement n. 779257) H2020 EU projects (to AF as full partner) is acknowledged; support from the National Natural Science Foundation of China No. 31800765 (to MF) is also acknowledged. AF is Chair of the Genetic Task of the European Reference Network 26 Euro-NMD (www.ern.euro-nmd.eu). We wish to acknowledge the work of the DMD gene database curator, Prof. Johan denDunnen (LOVD, www.lovd.nl)]

 [The funders had no role in study design, data collection and analysis, decision to publish, or preparation of the manuscript.]

Reviewers' comments:

Reviewer's Responses to Questions

**Comments to the Author**

1. Is the manuscript technically sound, and do the data support the conclusions?

Reviewer #1: Yes

Reviewer #2: Yes

2. Has the statistical analysis been performed appropriately and rigorously? 

Reviewer #1: Yes

Reviewer #2: Yes

3. Have the authors made all data underlying the findings in their manuscript fully available?

Reviewer #1: Yes

Reviewer #2: Yes

4. Is the manuscript presented in an intelligible fashion and written in standard English?

Reviewer #1: Yes

Reviewer #2: Yes

5. Review Comments to the Author

Reviewer #1: The manuscript is generally adequately well written. The authors structure it in a scientifically sound way, and the data are analysed adequately. The nature of the manuscript prohibits it from being broken down in more pieces so it can be more easily digested, therefore It is hard to suggest an improvement in terms of readability. The manuscript unfolds an important issue, and supports it in a thorough manner, providing some novel insights, perspective and future directions concerning disease related genes and codon usage bias. It should be accepted after minor revisions.

Suggestions

1.Line 65, “genes casing human rare diseases”

2. Line 93 “an extreme CUB was identified in both human and chimpanzee compared to other mammals, suggesting its evolutionary meaning in processes plays an exclusive role in a specific eukaryotic lineage” : This part could be rephrased making it clearer

3. Concerning this part :

95 “A large consensus exists about the concept that the choice of a synonymous codon does affect… for all genes [8]”.

- It is an important statement that is being made in the manuscript, about the consensus that exists around “choice of a synonymous codon does affect protein translation efficiency, expression level, structure, and function, a notion that has prompted”.

-This statement, is not adequately represented by references, hence some more references would further develop it.

4. -At the results section “CUB fingerprint across mammals” it is stated from line 277 until line 284 that

“…no clear tissue behavior or even specific gene-related clustering can be observed. This finding further supports that CU values show an mammals-related but with important gene-specific differences, which contribute to generate the CUB fingerprints”

-Although this statement can stand on its own, if it could be commented along with the statement at lines 92-94 concerning chimps and humans…

92- “an extreme CUB was identified in both human and chimpanzee compared to other 93mammals, suggesting its evolutionary meaning in processes plays an exclusive role in a 94specific eukaryotic lineage”

5. In the discussion part, at lines 403 to 404, where a hypothesis is presented concerning the reason of CUB , it is stated that positive selection of specific codons could putatively be the reason for this fact. Since it is an aspect of the “story” the manuscript unfolds, maybe these codons should be mentioned along with a more specific hypothesis, as it is very nicely done at the following lines concerning muscle tissue and the stop codon.

Reviewer #2: n the submitted manuscript , Dr. Rachele Ross et al try to use a new approach to analyze CUB in genes causing human disease-causing genes and non-disease-causing genes to understand connection between CUB and gene mutational diseases. In Particular, authors calculated CU values in tissue-specific and mostly expressed genes in three tissues and analyzed CU values between DC and NDC genes across 15 mammal species. In the end, they concluded that their approach described in this manuscript may be used to uncover connection between codon usage and rare diseases.

Comments:

1. Authors only focused on tissue-specific genes. I'm wondering whether authors also calculated CU values in housekeeping genes related to disease( disease-causing genes) vs genes not related to disease ?

2. Why did authors analyze CU values in 15 mammalian species ? Do these species develop the same diseases ? What is the scientific background regarding it ?

3. Please add some background information into the abstract.

6. PLOS authors have the option to publish the peer review history of their article (what does this mean?). If published, this will include your full peer review and any attached files.

Reviewer #1: No

Reviewer #2: No

---

## [Author Response · Author response to Decision Letter 0]

19 Feb 2022

We thank the referee for the useful comments and here below we provided our replies:

In response to Referee 1:

1.Line 65, “genes casing human rare diseases”: Amended in the manuscript text

2. Line 93 “an extreme CUB was identified in both human and chimpanzee compared to other mammals, suggesting its evolutionary meaning in processes plays an exclusive role in a specific eukaryotic lineage” : This part could be rephrased making it clearer: Amended in the manuscript text

3. Concerning this part :

95 “A large consensus exists about the concept that the choice of a synonymous codon does affect… for all genes [8]”. - It is an important statement that is being made in the manuscript, about the consensus that exists around “choice of a synonymous codon does affect protein translation efficiency, expression level, structure, and function, a notion that has prompted”.-This statement, is not adequately represented by references, hence some more references would further develop it. We added more reference to support the importance of the statement we reported in the manuscript.

4. -At the results section “CUB fingerprint across mammals” it is stated from line 277 until line 284 that“…no clear tissue behavior or even specific gene-related clustering can be observed. This finding further supports that CU values show an mammals-related but with important gene-specific differences, which contribute to generate the CUB fingerprints”

Although this statement can stand on its own, if it could be commented along with the statement at lines 92-94 concerning chimps and humans…

92- “an extreme CUB was identified in both human and chimpanzee compared to other 93mammals, suggesting its evolutionary meaning in processes plays an exclusive role in a 94specific eukaryotic lineage” Amended in the manuscript text

5. In the discussion part, at lines 403 to 404, where a hypothesis is presented concerning the reason of CUB , it is stated that positive selection of specific codons could putatively be the reason for this fact. Since it is an aspect of the “story” the manuscript unfolds, maybe these codons should be mentioned along with a more specific hypothesis, as it is very nicely done at the following lines concerning muscle tissue and the stop codon. Amended in the manuscript text

In response to Referee 2

1-Authors only focused on tissue-specific genes. I'm wondering whether authors also calculated CU values in housekeeping genes related to disease ( disease-causing genes) vs genes not related to disease

We apologies to the reviewer for having not explained enough clearly our approach. Our rationale was to consider tissue-specific genes only, either for non-disease causing (“control”) and disease causing genes and to compare them. Therefore, in this proof-of-principle study, we did not include housekeeping genes. Indeed, we definitively believe that housekeeping genes, almost ubiquitously expressed, would deserve to be studied by the same approach. This is something we will certainly do in further studies. In this paper we were focused on tissue specificity, since having genes with same expression site/level facilitated CU values calculation and comparison between gene groups.

Why did authors analyze CU values in 15 mammalian species ? Do these species develop the same diseases ? What is the scientific background regarding it ? 

Thanks for this question which allows to clarify the evolutionary analyses. In order to make more robust the identification of rare codons, we checked conservation of selected genes (both disease and non-disease causing) across mammals. We limited our selection to mammals to avoid to get i) too many bias due to the large specie variability, ii) to compare homo sapiens genes to species evolutionarily close (mammals and non-human primates). Regarding disease propensity, we disregarded wether species may develop human-like diseases, for two main reasons: this occurs very rarely (the so called natural animal disease models), and we were interested in comparing human genes only, since certainly related to diseases. Indeed, CUB was conserved among species, with however some difference in the heatmaps observed in disease-causing genes, especially those related to muscle diseases.

Concluding, analyzing CU values across mammals provided a further support of the different CUB behavior (we found conserved among species) between disease and non-disease genes. We have modified and added a sentence (Introduction) to clarify this concept.

---

## [Decision Letter · Decision Letter 1]

3 Mar 2022

Calculating and comparing codon usage values in rare disease genes highlights codon clustering with disease-and tissue- specific hierarchy

PONE-D-21-32048R1

Dear Dr. Ferlini,

We’re pleased to inform you that your manuscript has been judged scientifically suitable for publication and will be formally accepted for publication once it meets all outstanding technical requirements.

Kind regards,

Yanbin Yin

Academic Editor

PLOS ONE

Additional Editor Comments (optional):

Reviewers' comments:

Reviewer's Responses to Questions

**Comments to the Author**

1. If the authors have adequately addressed your comments raised in a previous round of review and you feel that this manuscript is now acceptable for publication, you may indicate that here to bypass the “Comments to the Author” section, enter your conflict of interest statement in the “Confidential to Editor” section, and submit your "Accept" recommendation.

Reviewer #1: All comments have been addressed

2. Is the manuscript technically sound, and do the data support the conclusions?

Reviewer #1: Yes

3. Has the statistical analysis been performed appropriately and rigorously? 

Reviewer #1: Yes

4. Have the authors made all data underlying the findings in their manuscript fully available?

Reviewer #1: Yes

5. Is the manuscript presented in an intelligible fashion and written in standard English?

Reviewer #1: Yes

6. Review Comments to the Author

Reviewer #1: The authors improved the manuscript implementing the suggestions/recommendations. Therefore my suggestion is that the manuscript be accepted.

7. PLOS authors have the option to publish the peer review history of their article (what does this mean?). If published, this will include your full peer review and any attached files.

Reviewer #1: No

---

## [Editor Report · Acceptance letter]

17 Mar 2022

PONE-D-21-32048R1 

Calculating and comparing codon usage values in rare disease genes highlights codon clustering with disease-and tissue- specific hierarchy 

Dear Dr. Ferlini:

I'm pleased to inform you that your manuscript has been deemed suitable for publication in PLOS ONE. Congratulations! Your manuscript is now with our production department. 

Kind regards, 

on behalf of

Dr. Yanbin Yin 

Academic Editor

PLOS ONE